# Inactivation of Human Norovirus GII.4 and *Vibrio parahaemolyticus* in the Sea Squirt (*Halocynthia roretzi*) by Floating Electrode-Dielectric Barrier Discharge Plasma

**DOI:** 10.3390/foods12051030

**Published:** 2023-02-28

**Authors:** Min Gyu Song, So Hee Kim, Eun Bi Jeon, Kwang Soo Ha, Sung Rae Cho, Yeoun Joong Jung, Eun Ha Choi, Jun Sup Lim, Jinsung Choi, Shin Young Park

**Affiliations:** 1Department of Seafood Science and Technology, Gyeongsang National University, Tongyeong 53064, Republic of Korea; 2Southeast Sea Fisheries Research Institute, National Institute of Fisheries Science, Tongyeong 53085, Republic of Korea; 3Food Safety and Processing Research Division, National Institute of Fisheries Science, Busan 46083, Republic of Korea; 4Department of Electrical and Biological Physics, Plasma Bioscience Research Center, Kwangwoon University, Seoul 01987, Republic of Korea

**Keywords:** sea squirt, FE-DBD plasma, human norovirus GII.4, *Vibrio parahaemolyticus*, disinfection, food pathogen, quality

## Abstract

Human norovirus (HNoV) GII.4 and *Vibrio parahaemolyticus* may be found in sea squirts. Antimicrobial effects of floating electrode-dielectric barrier discharge (FE-DBD) plasma (5–75 min, N_2_ 1.5 m/s, 1.1 kV, 43 kHz) treatment were examined. HNoV GII.4 decreased by 0.11–1.29 log copy/μL with increasing duration of treatment time, and further by 0.34 log copy/μL when propidium monoazide (PMA) treatment was added to distinguish infectious viruses. The decimal reduction time (D_1_) of non-PMA and PMA-treated HNoV GII.4 by first-order kinetics were 61.7 (R^2^ = 0.97) and 58.8 (R^2^ = 0.92) min, respectively. *V. parahaemolyticus* decreased by 0.16–1.5 log CFU/g as treatment duration increased. The D_1_ for *V. parahaemolyticus* by first-order kinetics was 65.36 (R^2^ = 0.90) min. Volatile basic nitrogen showed no significant difference from the control until 15 min of FE-DBD plasma treatment, increasing after 30 min. The pH did not differ significantly from the control by 45–60 min, and Hunter color in “L” (lightness), “a” (redness), and “b” (yellowness) values reduced significantly as treatment duration increased. Textures appeared to be individual differences but were not changed by treatment. Therefore, this study suggests that FE-DBD plasma has the potential to serve as a new antimicrobial to foster safer consumption of raw sea squirts.

## 1. Introduction

The sea squirt (*Halocynthia roretzi*) is an attached marine animal that belongs to the phylum Chordata. Since ancient times, it has been enjoyed as food because of its unique smell and taste [1]. Sea squirts consume plankton as food. They attach to rocks in the sea at a water temperature of 5 to 24 °C and depth of 6 to 20 m. In Korea, sea squirts are produced mainly on the south and east coasts [2]. The sea squirt production period is limited from late spring to summer; hence, it is often made into traditional Korean salted and fermented food, *jeotgal,* for long-term storage [3]. However, due to economic growth and industrialization, pollution along the coast close to the land is accelerating [4]. In addition, most sea squirts are consumed, raw without heat treatment, therefore, contamination with harmful microorganisms can cause food poisoning.

Human norovirus (HNoV) is a typical virus that causes food poisoning. Norovirus is a circular RNA virus belonging to the *Caliciviridae* family and has an incubation period of 24–48 h. Symptoms of infections begin with typical gastroenteritis symptoms, such as abdominal pain, diarrhea, fever, and nausea. Genogroups of norovirus are largely classified into five categories, of which GI, II and IV are known to infect humans [5]. HNoV food poisoning is transmitted via a fecal-oral route, in which food is contaminated by the feces of infected patients, and when those contaminated foods are consumed, illness occurs [6]. HNoV occurs especially in shellfish such as oysters and sea squirts that are produced on contaminated coasts [7,8,9], and these can be consumed raw, so attention should be paid to safety.

*Vibrio parahaemolyticus* is a gram-negative halophilic bacterium that primarily lives in seawater. Infection occurs by ingesting raw or insufficiently cooked fish and shellfish contaminated with these bacteria. These are food poisoning bacteria that cause acute gastritis symptoms, mainly accompanied by abdominal pain, diarrhea, vomiting, chills, and mild fever [10]. Symptoms usually occur after an incubation period of 4–96 h. Incidents of food poisoning caused by *Vibrio parahaemolyticus* mainly occur in the summer between June and October [11,12,13]. In Korea, 30.5% of raw fish products, including sea squirts, are reported to be infected with *V*. *parahaemolyticus* [11,12,13,14].

Plasma is the fourth state of matter. It can be used for non-thermal disinfection which can be applied to various biomedical applications [15]. In the generated plasma, ions and electrons are separated, and reactive species with high chemical reactivity and ozone are formed [16]. Active oxygen species (ROS) and active nitrogen species (RNS) produced by plasma-generated gas ionization exhibit antibacterial effects through direct and specific attacks on microbial cell envelopes and intracellular components [17]. Dielectric barrier discharge (DBD) plasma technology is a solution to problems related to thermal disinfection using plasma. The antibacterial effect of DBD plasma has the advantages of low-temperature treatment, minimal nutrient destruction, and texture maintenance [18]. In DBD plasma, the current is limited by using a dielectric covering on the two electrodes and discharging the current between the two electrodes at atmospheric pressure to generate plasma, which exhibits a disinfection effect [19]. Recently, floating electrode-dielectric barrier discharge (FE-DBD) plasma, which further enhances DBD plasma, has been developed. The FE-DBD plasma grounds one of the two electrodes, which is directly discharged through a sample (food) to generate ROS and RNS on the surface of the sample. It has stronger disinfection power [20].

Studies have investigated the effect of reducing harmful microorganisms using FE-DBD plasma [21,22,23]; however, the effect on reducing HNoV and *V*. *parahaemolyticus* among sea squirts is still insufficiently understood. Therefore, in this study, the effect of reducing HNoV GII.4 and *V*. *parahaemolyticus* in sea squirt by FE-DBD plasma treatment and the resulting quality change were investigated.

## 2. Materials and Methods

### 2.1. HNoV GII.4 Preparation

HNoV GII.4 used in this study was isolated from patients with gastroenteritis symptoms caused by norovirus at the Gyeonggi Institute of Health and Environment (GIHE; Gyeonggido, Republic of Korea) in 2019. After confirming the genotype of HNoV, it was stored in the Waterborne Virus Bank (WAVA; Seoul, Republic of Korea). The HNoV GII.4 used in this experiment was purchased from WAVA and delivered frozen. After purchase, it was manufactured from stock containing 500 μL phosphate buffer solution (PBS; pH 7.2), stored in a −80 °C freezer for further experimental use.

### 2.2. Vibrio parahaemolyticus Preparation

*V. parahaemolyticus* (ATCC 27969) was used in the experiments. Stock cultures were stored at −80 °C in tryptic soy broth (TSB; Difco Laboratories, Detroit, MI, USA) containing 30% glycerol. *Vibrio parahaemolyticus* was cultivated in TSB containing 2.5% NaCl. This process was performed twice for the bacterial activity. The strain (10 μm) was incubated in 5 mL TSB for 24 h at 37 °C and centrifuged at 4695× *g* for 10 min at 4 °C (SUPRA22K, Daejeon Hanil Science Industry). After centrifugation, the TSB was removed and the pellets were mixed in 9 mL of sterile 0.85% NaCl.

### 2.3. FE-DBD Plasma Treatment of HNoV GII.4 and Vibrio parahaemolyticus in Sea Squirt

Frozen sea squirt used as a sample for the research was purchased at a local market in Tongyeong, Korea. Immediately after purchase, it was stored in a freezer (−18 °C) and the experiment was conducted within 48 h. The frozen sea squirt was completely defrosted and the intestines were collected with sterile tweezers and scissors and then homogenized with a homogenizer (stirrer, Daihan Scientific Co., Wonju, Republic of Korea). Homogenized intestines were divided into 3 g portions and placed in Petri dishes. The Petri dishes containing samples were inoculated with 10 μL (2.55 log copy/μL) HNoV GII.4 and 100 μL (4.16 log CFU/g) *V*. *parahaemolyticus,* respectively, and were placed into a clean bench (CHC LabCo. Ltd., Daejeon, Republic of Korea) for 1 h.

The FE-DBD plasma device was screen-printed with a high-voltage electrode on glass with a thickness of 10 μm. The thickness was 7 mm, and the genetic material composed of SiO_2_ was also screen-printed to a thickness of 100 μm. The operating voltage was supported by inverters that generated 47 kHz sine waves with an amplitude of 2.8 kV. Plasma was generated between the surface of the sample, which acted as a virtual ground using nitrogen, and the glass under the electric electrode. During the procedure, a 1.5 L flow rate per min and the distance between the plasma release electrode and the sample was maintained at 3 mm and treated for 5, 15, 30, 45, 60 and 75 min. The electrical voltage and current characteristics of the FE-DBD were measured using a high-voltage probe (P6015A, Tektronix, Beaverton, OR, USA) and a pickup probe (P6021A, Tektronix, Beaverton, OR, USA), respectively. The plasma discharge mainly occurred at 1 kV, and the peak discharge was measured at 16 mA. The electric scattering power was measured at 0.55 W. The root mean square value (RMS) voltage and current were measured at 2.0 kV and 13.5 mA, respectively. During the experiment, we used samples as negative control without any treatment of FE-DBD plasma for each contact time.

### 2.4. Propidium Monoazide (PMA) Treatment in HNoV GII.4

For PMA treatment, samples inoculated with HNoV GII.4 were immediately mixed with 200 μM PMA (Biotium, Hayward, CA, USA) and placed at room temperature for 5 min to ensure sufficient dye penetration. Afterwards, to photoactivate the dye, the sample was exposed to 40 W LED light (Dinebio, Seongnam, Republic of Korea) with a wavelength of 460 nm at room temperature for 20 min. A control treated with PMA and not exposed to halogen light was included to determine whether the dye treatment interfered with virus detection. Finally, virus samples were differentially detected as potentially infectious or non-infectious virus particles of HNoV, according to optimized PMA pretreatment prior to RT-qPCR analysis.

### 2.5. RNA Extraction

An RNeasy mini kit (Qiagen, Hilden, Germany) was used for RNA extraction of HNoV GII.4 and the experiment was conducted according to the manufacturer’s instructions. Proteinase K extraction activity was processed according to the ISO 15216-1:2017 method. Proteinase K (Sigma, St. Louis, MO, USA) was added to the FE-DBD plasma-treated sea squirt, shuffled in an incubator (37 °C) for 1 h, and then deactivated in a thermostat (60 °C) for 15 min. In addition, centrifugation (5400 rpm, 4 °C) was performed for 10 min using a centrifuge (SUPRA22K, Hanil Science Industrial Co., Gimpo, Republic of Korea), and a clear solution (approximately 3.0 mL) in the upper layer was collected in a sterile conical tube. This solution was stored in a freezer at −80 °C and used for the detection and quantification analysis of HNoV GII.4.

### 2.6. Quantitative Analysis of HNoV GII.4 Infectivity by RT-qPCR

Reverse transcription of cDNA was performed as described by Kageyama et al. [24]. In order to amplify the gene of HNoV GII.4, RNase free water, enzyme mix (5 units/μL), 5X RT-PCR buffer, 10 mM dNTP, and 10 μM primer (forward and reverse extraction) were added to amplify the gene of HNV GII.4, and then 10 μM primer (forward and reverse NA) RT-qPCR A TP800-thermal cycler dice real-time system (TaKaRa) device was used for quantitative analysis of real-time reverse polymerase chain reaction using real time reverse transcription-quantitative polymerase chain reaction (RT-PCR).

Primers and probes were designed to fit the ORF-1 and ORF-2 overlapping regions of HNoV GII.4, increasing sensitivity and specificity; the base sequences of the primer and probe are shown in Table 1. In addition, RNA from HNoV GII.4 was used as a positive control and RNase-free water was used as a negative control.

### 2.7. Quantitative Analysis of V. parahaemolyticus by Standard Plate Count

The FE-DBD plasma-treated *V*. *parahaemolyticus* samples were homogenized in stomacher (Easy Mix, AES Chemunex, Bruz, BRE, France) by adding 0.85% sterile NaCl solution to a sterile bag. The homogenized solution was diluted with 9 mL of 0.85% sterile NaCl solution. The diluted sample (1 mL) was placed in a Petri dish, mixed with 2.5% NaCl containing tryptic soy agar (Difco, Ditroit, MI, USA), and cultured at 37 °C for 24 h, and a cluster of 15–300 was calculated per 1 mL of the sample solution.

### 2.8. Volatile Basic Nitrogen (VBN)

VBN was measured by a microdiffusion analysis method using Conway units. Distilled water (25 mL) and 5 mL of 20% trichloroacetic acid (TCA) were added to 5 g of the sample, mixed well, leached, and filtered for 30 min, and then, a 2% TCA solution was added to the filtrate to use a Conway unit container as a test solution. The test solution was then added to the outer chamber of the Conway unit. Next, 1 mL of 0.01 N boric acid and 100 μL of a Conway reagent (0.066% (w/v) methyl red and 0.066% (w/v) bromocresol in ethanol) were added to the inner chamber of the Conway unit, and 1 mL of 50% (w/v) carbonate was added to the other chamber of the unit. The unit was then sealed and slowly stirred in a horizontal direction to mix reagents in the outer chamber and incubated at 37 °C for 120 min. After incubation, the inner chamber of the Conway unit was titrated with 0.02 N sulfuric acid.

### 2.9. Hunter Color and pH

The color of the sea squirt was measured using a color meter (UltraScan PRO, Hunterlab, Reston, VA, USA) after FE-DBD plasma treatment and calibrated to the original value of the standard plate (‘L’ = 98.48, ‘a’ = 0.14 and ‘b’ = 0.41). The color was measured through the 6 mm aperture of a color meter using a D65 illuminant. Values were represented by three coordinate values: ‘L’ (brightness +, darkness −), ‘a’ (red +, green −), and ‘b’ (yellowness +, blue −) depending on the Hunter color.

For pH measurement, 3 g of FE-DBD plasma-treated (0, 5, 15, 30, 45, 60 and 75 min) sea squirt and 27 mL of sterile distilled water were mixed. Thereafter, the pH was homogenized with a stomacher (Easy Mix, AES Chemunex, French Ren) for 3 min and the pH value was measured three times using a pH meter (Orion Star A211, Thermo Scientific, Troy, MI, USA).

### 2.10. Texture

The texture profile analysis (TPA) of sea squirts was modified and set using published parameters [25,26]. The mechanical properties of sea squirts were obtained from FE-DBD plasma samples treated for each time and evaluated using a CT3 texture analyzer (Middleboro, MA, USA). The TPA characteristics evaluated were hardness (g/cm^2^) and chewiness. A spherical stainless-steel probe TA18 (diameter 12.7 mm) was used to perform two consecutive compression cycles separated after 5 s with the compression system set to reach 50% deformation. The probe was set to a trigger force of 5.0 g at a constant speed of 0.5 mm/s. To eliminate the influence of the weight or size of the sea squirts, the texture profile of animals of the same size was recorded within an error range of ±1 g.

### 2.11. Statistical Analysis

All measurements were recorded in three independent experiments, and each test was performed using three samples. One-way analysis of variance (ANOVA), and Duncan’s multi-range tests were performed using the statistical packages for social science (SPSS) version 25.0 (SPSS Inc., Chicago, IL, USA). Statistical analysis was performed to determine significant differences between the mean values of viruses and bacteria. Paired t-tests were performed to evaluate the statistical significance of the differences in PMA treatment reduction using SPSS software (log CFU/g). Statistical significance was determined at the 5% probability level (*p* < 0.05).

## 3. Results

### 3.1. Reduction of HNoV GII.4

Table 2 showed the trend of HNoV GII.4 in sea squirts inoculated with HNoV GII.4 according to FE-DBD plasma treatment (0, 5, 15, 30, 45, 60 and 75 min). The initial HNoV GII.4 titer of the sample without FE-DBD plasma treatment was 2.55 log copy/μL. The results were analyzed by comparing the non-PMA and PMA-treated viruses. First, samples that were not PMA-treated showed significant differences (*p* < 0.05) in the reduction effect as the treatment time increased, except for 5–15 min of treatment time. Compared to the control, the log reduction effect of treatment time was as follows: 5 min treatment (0.11 log reduction), 15 min treatment (0.26 log reduction), 30 min treatment (0.35 log reduction), 45 min treatment (0.57 log reduction), 60 min treatment (0.95 log reduction), and 75 min treatment (1.29 log reduction). Second, the PMA-treated samples showed a significant difference (*p* < 0.05) in the remaining treatment time except for 15–30 min. The differences between the PMA and non-PMA-treated samples were compared using t-tests and the results were follows: 5 min treatment (2.44 − 2.07 = 0.37), 15 min treatment (2.29 − 1.92 = 0.37), 30 min treatment (2.20 − 1.79 = 0.41), 45 min treatment (1.98 − 1.65 = 0.33), 60 min treatment (1.60 − 1.39 = 0.21), and 75 min treatment (1.26 − 0.96 = 0.30). The D-values of the non-PMA and PMA-treated samples calculated by the first kinematic model were 61.7 and 58.4 min, respectively, and R^2^ were 0.97 and 0.92, respectively (Figure 1).

### 3.2. Reduction of V. parahaemolyticus

Table 3 shows the reduction of *V. parahaemolyticus* in sea squirts using FE-DBD plasma. In the untreated controls, the microbial titer was 4.16 log CFU/g, which tended to decrease with increasing treatment time (5–75 min) (*p* < 0.05). FE-DBD plasma for 5, 15, 30, 45, 60, and 75 min resulted in 4.00 (0.16 log reduction), 3.98 (0.18 log reduction), 3,76 (0.4 log reduction), 3.54 (0.62 log reduction), 3.41 (0.75 log reduction), and 2.66 (1.5 log reduction) log CFU/g, respectively. Excluding the 5–15 min treatment, significant differences (*p* < 0.05) were observed in the reduction effect as the treatment time increased. The D-value of *V*. *parahaemolyticus*, calculated using the first-order kinetics model, was 58.9 min and R^2^ was 0.90 (Figure 2).

### 3.3. Effect of DBD Plasma Treatment on D_1_ Values of HNoV GII.4, HNoV GII.4 with PMA and V. parahaemolyticus in Sea Squirt

D_1_ values were obtained using a first-order kinematic model based on the survival curves of HNoV GII.4, HNoV GII.4 with PMA and *V. parahaemolyticus* generated for samples treated with various DBD plasma treatment times. In the case of the dynamics for the inactivation of microorganisms, the decimal reduction time of the log linear kinetic model is widely accepted. The D_1_ values of HNoV GII.4 and HNoV GII.4 with PMA were 61.96 and 58.68 min, respectively, and the R^2^ values were 0.97, 0.92. The D_1_ value of Vibrio was 58.70 min and the R^2^ value was 0.90. This indicated that the log linear kinetic models for HNoV GII.4, HNoV GII.4 with PMA, and *V. parahaemolyticus* were suitable for determining the D_1_ value (Table 4).

### 3.4. Effect of FE-DBD Plasma on VBN and pH of Sea Squirt

The VBN and pH values are listed in (Table 5). VBN compared samples treated for 5–75 min with the control. The value was 8.3–12.4, and it increased significantly when the cell were treated for more than 30 min. The pH was also compared with the control for 5–75 min treated samples, and the values were at least 5.78, up to 5.94, with significant differences between the individuals.

### 3.5. Effect of FE-DBD Plasma on Hunter Color of Sea Squirt

Hunter color compared to the control not treated with FE-DBD plasma and the samples treated with FE-DBD plasma for 30 and 60 min are shown in Table 6. Compared with the control, “L” (Lightness), “a” (redness), and “b” (yellowness) showed significant differences as the FE-DBD plasma treatment time increased, and the value gradually decreased.

### 3.6. Effect of FE-DBD Plasma on Texture of Sea Squirt

Texture was analyzed to determine the effect of DBD plasma treatment on the sea squirt surface texture (Table 7). The hardness range obtained as a result of FE-DBD plasma treatment was 214–265 (g/cm^2^) and the chewing range was 11–19 (g/cm^2^); the difference between individuals was not associated with the treatment time.

## 4. Discussion

Increasing consumption of marine products is a global trend [27]. However, since marine products are exposed to seawater environments where various microorganisms exist and there is a possibility that they may be exposed to humid environments for a long time during production and distribution, the growth and metabolism of microorganisms may proceed actively [11]. In addition, hygiene management is very important because it is easy to change, and microbial contamination can easily occur in various channels such as distribution, processing, and consumption, such as post-change and rapid degradation of quality after fishing [28].

In Korea, sea squirt production was 22,833–38,248 metric tons per year as of 2015–2019, which make it one of the most industrially important aquaculture species in Korea [29]. However, most of them are caught by natural marine fishing; therefore, fish farms are distributed along the coast, and coastal waters are likely to be affected by various pollutants. In fact, Shin et al. [30] reported that the bacterial content was high on the coast of large rivers or densely populated areas and that fecal pollutants derived from land flow into the coast due to the occurrence of rainfall. According to a study by Vincent-Hubert et al. [31], many harmful microorganisms, such as HNoV and *Vibrio* spp., were detected off the coast, and Elbashir et al. [32] also reported that norovirus and *Vibrio* spp., which live on the coast, were sufficiently infectious to aquaculture. HNoV GII.4 and *V. parahaemolyticus* can occur frequently in marine products consumed without insufficient heating, and in Korea, where a lot of raw marine products are consumed, these cause infections in summer as well as during the four seasons. Globally, the problem of norovirus has been reported to result in social costs of approximately USD 60 billion annually [33]. The number of food poisoning cases caused by *V. parahaemolyticus* was approximately 855 in Korea between 2016 and 2020 [34], and the risk of food poisoning seems to be frequent even now when industrial development and food hygiene are taken more seriously. In preparation for such microbial contamination, food is subjected to appropriate disinfection treatment according to the characteristics of the food. The most important aspect of food disinfection is that there should be no change in quality. Excessive heat treatment may negatively affect the taste or texture of food, destroy nutrients, and cause discoloration [35]. Therefore, in this study, the effect of sterilizing HNoV GII.4 and *V. parahaemolyticus* in sea squirts was investigated using FE-DBD plasma, a non-thermal atmospheric plasma technology that did not significantly affect the quality of food, and quality analysis (VBN, pH, Hunter color, and texture) was also performed.

Unlike DBD plasma using two existing floating electrodes, FE-DBD plasma directly causes plasma on the sample surface using one floating electrode, showing strong disinfection ability. In the case of FE-DBD plasma treatment by HNoV GII.4 among sea squirts, it was found that there was a significant difference according to the processing time in this study, and the HNoV GII.4 decreased as the duration time increased, and up to 1.26 log copy/μL decreased. Similar to this study, the FE-DBD plasma treatment for HNoV GII.4 in salted clams reported by Jeon et al. [21] also showed reduction of viable HNoV GII.4 by FE-DBD treatment, and the FE-DBD plasma treatment was effective in deactivating HNoV GII.4. In addition, Csadek et al. [36] showed a reduction in RNA viruses of up to 3.4 and 1.4 log copy/μL in 15 min when treated with high-power and low-power cold plasma viruses, respectively. Filipic et al. [37] demonstrated the effectiveness of cold plasma treatment on the inactivation of viruses such as norovirus and hepatitis virus A. Most virus inactivation methods are based on attacking the capsid, a structure that encloses the virus [38]. Active oxygen and nitrogen species produced during plasma exposure cause cell wall erosion, cell membrane destruction, functional changes, capsid destruction or structural changes, and loss of infectivity [39].

RT-qPCR has significantly improved virus detection capabilities, is also widely used in norovirus detection, and has proven to be effective [40,41,42]. Although RT-qPCR has excellent functionality in virus detection and quantification, it is limited in that it cannot distinguish between living and dead viruses owing to the persistence of DNA and RNA [43]. Recently, to overcome these shortcomings, a dye called propidium monoazide (PMA) has been added to distinguish between infectious and non-infectious viruses. PMA infiltrates the damaged viral capsid when treated with fluorescent dyes and then photosynthesizes using UV light to covalently bind to damaged RNA, preventing further amplification of RNA reverse transcription and RT-qPCR, thereby enabling the distinction between living viruses and dead viruses. However, PMA treatment is not effective for all virus inactivation methods because it depends on method of inactivation of the target virus [44]. In this study, when comparing non-PMA-treated samples with PMA-treated samples, the average of 0.34 log copy/μL was further reduced, and PMA treatment was found to be effective in detecting living viruses. Kim et al. [45] recently reported that PMA treatment helped detect live viruses in HNoV in oysters irradiated with electron beams, and Choi et al. [46] also found that up to 0.92 log copy/μL was reduced in DBD plasma treatment in HNoV in oysters, and the plasma treatment caused sufficient damage to the capsules of the virus.

Active species from plasma can be used to disinfect most microorganisms, including viruses, bacteria [47], and fungi [48], resulting in cell wall erosion, cell membrane destruction, functional changes, and DNA damage. In this study, when the reduction of *V. parahaemolyticus* in sea squirt through FE-DBD plasma treatment was observed, there was a significant viable reduction as the treatment time increased, and a maximum reduction of 1.5 log CFU/g was observed at 75 min. Kim et al. [49] recently confirmed the disinfection effect of DBD plasma treatment on *Escherichia coli* and *V. parahaemolyticus* which reduced *V. parahaemolyticus* by up to 1.3 log CFU/g after 60 min of treatment, similar to the result of this study.

In this study, VBN, pH, Hunter color, and texture were measured using quality parameters according to the FE-DBD duration. VBN is used to determine the degree of decomposition of protein-rich foods, such as meat and seafood [50]. If VBN is 5–10 mg/100 g, it is very fresh, and if VBN is 15–25 mg/100 g, the level above the normal line is usually determined to be corrupt [51]. In this study, there was no significant difference compared to the control until the 15 min treatment, but there was a significant difference from 30 min onwards, and the VBN content increased with time. In a study by Shin et al. [52], the VBN level of processed sea squirt products increased over time, and in a study by Oh [53], the VBN content of marine products increased over time. In this study, as the FE-DBD plasma treatment period increased, the VBN content increased accordingly, so it was judged that the treatment should be conducted while maintaining a low temperature. However, although the VBN content increased, it is likely that there will be no problems in intake and processing because the values remain located between very fresh and normal levels. No significant differences between the pH of the control and FE-DBD plasma-treated samples were observed at 45 min and 60 min. Considering that there were no significant differences between the samples with the longest treatment time and the control, observed differences are likely to be differences between individuals rather than a change associated with duration of FE-DBD plasma treatment. Choi et al. [39] reported a DBD plasma study on oysters where there was no pH change due to an increase in duration of plasma treatment. As the duration of FE-DBD plasma treatment increased, there was a significant difference in Hunter color. Abdi et al. [22] reported that when red pepper was treated with low-temperature plasma for 20 min, there was a significant decrease in the control. In addition, a study by Choi et al. [46] showed that the “L”, “a”, and “b” values were significantly reduced in frozen pork treated with corona discharge plasma jet. Owing to the characteristics of FE-DBD plasma, which is composed of one electrode, it is likely that the color change was caused by direct reaction of the plasma on the sample surface. Texture seems to be an individual characteristic, not a difference according to processing duration, and it is likely that the FE-DBD plasma does not cause a difference in texture. Choi et al. [54] reported that there was no significant difference in the change in the texture of oysters corresponding to the duration of DBD plasma treatment. Because of the nature of the sea squirt used in this study, it was likely that the observed results were due to severe differences between individuals in factors such as shape and thickness, which can affect texture. However, as shown in previous studies, plasma treatment is not expected to have a significant effect on texture.

## 5. Conclusions

The current study demonstrated that 1.29 and 1.50 log reductions of HuNoV and *V. parahaemolyticus* in the sea squirts were achieved following FE-DBD plasma treatment for 60 min. These results suggest that PMA/RT-qPCR may be useful in detecting HuNoV infectivity following FE-DBD plasma treatment for an extended exposure time. Based on first-order kinetics (R^2^ = 0.92 and 0.90, respectively, for HuNoV and *V. parahaemolyticus*) following the FE-DBD plasma treatment of sea squirt, there was no significant difference in pH between the control group and the 45–60 min treatment time. Hunter color values of “L”, “a”, and “b” decreased as the FE-DBD plasma treatment period increased. Texture was not significantly different under FE-DBD plasma treatment. The results also suggest that FE-DBD plasma could minimize changes in quality, which may be a potential new physical method for improving the safety of sea-squirt consumption by reducing pathogenic microorganisms in sea squirts.

## Figures and Tables

**Figure 1 foods-12-01030-f001:**
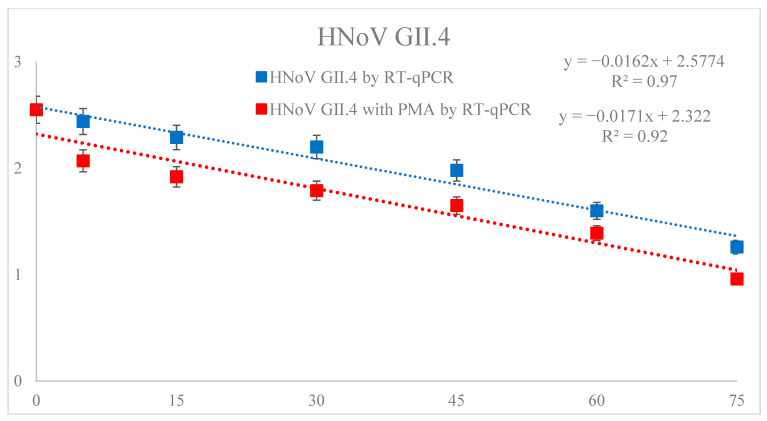
Fitted survival curves of HNoV GII.4 infectivity from FE-DBD plasma-treated sea squirt using the first-order kinetic model. R^2^ = correlation coefficient (a higher R^2^ value indicates a better fit to the data). Data were represented as mean ± standard deviation of three independent replicates.

**Figure 2 foods-12-01030-f002:**
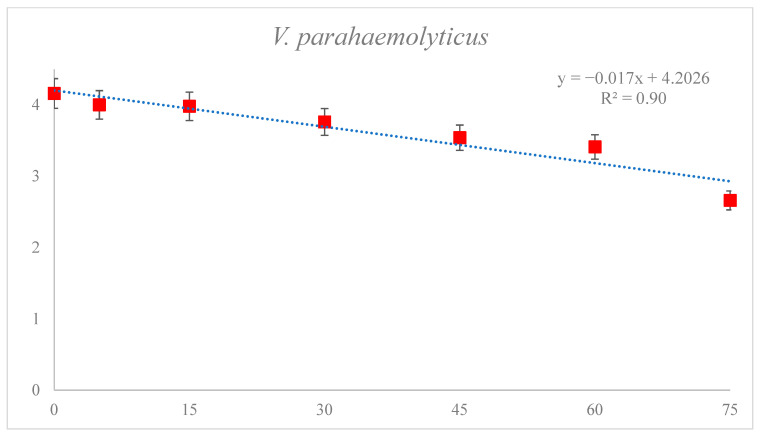
Fitted survival curves of *V. parahaemolyticus* infectivity from FE-DBD plasma-treated sea squirt using the first-order kinetic model. R^2^ = correlation coefficient (a higher R^2^ value indicates a better fit to the data). Data were represented as mean ± standard deviation of three independent replicates.

**Table 1 foods-12-01030-t001:** Sequence of primers and probe for RT-qPCR to quantitate HNoV GII.4.

Genotype	Type	Component	Sequence (5′→3′)
GII	Primer	COG1F	5′-CAR GAR BCN ATG TTY AGR TGG ATG AG–3′
		COG2R	5′-TCG ACG CCA TCT TCA TTC ACA-3′
	Probe	RING2	5′-TGG GAG GGC GAT CGC AAT CT-3′

**Table 2 foods-12-01030-t002:** Effect of FE-DBD plasma treatment against HNoV GII.4 spiked on sea squirt.

FE-DBD Plasma (min)	Non-PMA/RT-qPCR	PMA/RT-qPCR	Before/After Using PMA to HNoV Reduction Difference (log Copy Number/µL)
	log Copy Number/µL	log Copy Number/µL	
0	2.55 ± 0.16 ^a^	2.55 ± 0.16 ^a^	
5	2.44 ± 0.07 ^Aa^	2.07 ± 0.04 ^Bb^	(2.44 − 2.07) = 0.37
15	2.29 ± 0.03 ^Ab^	1.92 ± 0.09 ^Bc^	(2.29 − 1.92) = 0.37
30	2.20 ± 0.01 ^Ab^	1.79 ± 0.01 ^Bc^	(2.20 − 1.79) = 0.41
45	1.98 ± 0.01 ^Ac^	1.65 ± 0.00 ^Bd^	(1.98 − 1.65) = 0.33
60	1.60 ± 0.02 ^Ad^	1.39 ± 0.06 ^Be^	(1.60 − 1.39) = 0.21
75	1.26 ± 0.07 ^Ae^	0.96 ± 0.06 ^Bf^	(1.26 − 0.96) = 0.30

Different letters within the same column (a–f) indicate significant differences (*p* < 0.05) by Duncan’s multiple range test. Different letters within the same row (A,B) indicate significant differences (*p* < 0.05) between non-treated and PMA-treated samples by student *t*-test. The data represent as mean ± standard deviation (SD). Number of replicates (n = 3).

**Table 3 foods-12-01030-t003:** Effect of FE-DBD plasma treatment against *V*. *parahaemoyticus* incubated sea squirt.

FE-DBD Plasma (min)	*V*. *parahaemolyticus*
	log CFU/g
0	4.16 ± 0.03 ^a^
5	4.00 ± 0.06 ^b^
15	3.98 ± 0.03 ^b^
30	3.76 ± 0.08 ^c^
45	3.54 ± 0.08 ^d^
60	3.41 ± 0.01 ^e^
75	2.66 ± 0.05 ^f^

Different letters within the same column (a–f) indicate significant differences (*p* < 0.05) by Duncan’s multiple range test. The data represent as mean ± standard deviation (SD). Number of replicates (n = 3).

**Table 4 foods-12-01030-t004:** Effect of FE-DBD plasma treatment on D-values of HNoV GII.4, HNoV GII.4 with PMA and *V. parahaemolyticus*, and reduction by first-order kinetics model in the sea squirt.

	Quantify	D-Value (min)	R^2^	y = −ax + b
HNoV GII.4	RT-qPCR	61.96 ± 3.28	0.97	y= −0.016x + 2.577
	PMA with RT-qPCR	58.68 ± 2.74	0.92	y= −0.017x + 2.322
*V. parahaemolyticus*	Standard plate count	58.70 ± 0.20	0.90	y= −0.017x + 4.202

The data present means for three samples with standard deviations (three samples/treatment). D-values, decimal of log reduction time. R^2^, correlation coefficient.

**Table 5 foods-12-01030-t005:** VBN and pH of sea squirt by FE-DBD plasma treatment.

FE-DBD Plasma (min)	VBN (mg/100 g)	pH
0	8.3 ± 0.1 ^c^	5.78 ± 0.04 ^c^
5	8.3 ± 0.1 ^c^	5.88 ± 0.04 ^ab^
15	8.2 ± 0.2 ^c^	5.88 ± 0.04 ^ab^
30	11.0 ± 0.1 ^b^	5.94 ± 0.05 ^a^
45	12.3 ± 0.1 ^a^	5.80 ± 0.00 ^c^
60	12.4 ± 0.2 ^a^	5.84 ± 0.05 ^bc^
75	12.4 ± 0.2 ^a^	5.94 ± 0.05 ^a^

Different letters within the same column (a–c) indicate significant differences (*p* < 0.05) by Duncan’s multiple range test.

**Table 6 foods-12-01030-t006:** Hunter color of sea squirt by FE-DBD plasma treatment.

FE-DBD Plasma (min)	Color		
	L’ Value	a’ Value	b’ Value
0	46.16 ± 0.02 ^a^	20.22 ± 0.02 ^a^	27.05 ± 0.01 ^a^
5	46.09 ± 0.01 ^b^	19.85 ± 0.01 ^b^	26.97 ± 0.01 ^b^
15	45.92 ± 0.01 ^c^	19.77 ± 0.02 ^c^	26.87 ± 0.01 ^c^
30	45.85 ± 0.01 ^d^	19.41 ± 0.02 ^d^	26.79 ± 0.01 ^d^
45	44.73 ± 0.02 ^e^	19.37 ± 0.01 ^e^	26.75 ± 0.01 ^e^
60	44.44 ± 0.01 ^f^	19.20 ± 0.01 ^f^	26.53 ± 0.01 ^f^
75	43.67 ± 0.01 ^g^	19.16 ± 0.01 ^g^	26.34 ± 0.01 ^g^

Different letters within the same column (a–g) indicate significant differences (*p* < 0.05) by Duncan’s multiple range test.

**Table 7 foods-12-01030-t007:** Texture of sea squirt by FE-DBD plasma treatment.

Time (min)	Texture	
	Hardness	Chewiness
0	233 ± 3.2	14 ± 0.5
5	261 ± 3.8	20 ± 0.8
15	255 ± 2.7	18 ± 0.9
30	234 ± 6.9	14 ± 0.5
45	240 ± 3.5	15 ± 0.7
60	250 ± 8.1	11 ± 0.3
75	265 ± 1.2	19 ± 0.1

Values are expressed as mean ± standard deviation (n = 3).

## Data Availability

Data is contained within the article.

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
