# Peer review of "Inactivation of Human Norovirus GII.4 and Vibrio parahaemolyticus in the Sea Squirt (Halocynthia roretzi) by Floating Electrode-Dielectric Barrier Discharge Plasma"

_foods, 2023, doi:10.3390/foods12051030_

Round 1

Reviewer 1 Report

The manuscript entitled “Inactivation of human norovirus G.4 and Vibrio parahaemolyticus in the sea squirt (Halocynthia roretzi) by floating electrode-dielectric barrier discharge plasma” analyzed the effect of floating electrode-dielectric barrier discharge plasma against human Norovirus GII.4 and V. parahaemolyticus to evaluate the antimicrobial effects of this treatment. The study seems interesting however all the tables with the data are completely missing despite being mentioned in the text. This does not allow a correct evaluation of the article. We therefore propose to resubmit the complete article before a final judgment.

In addition, there are some comments and mistakes that should be addressed.

Author Response

Thank you for the comment. The revised part is marked "Red" in the manuscript. There are answers in the REVIEWER 1 answer file. Thank you.

Reviewer 2 Report

Comments are provided in the attached file

Author Response

Thank you for the comment. The revised part is marked "Red" in the text. There are answers in the REVIEWER 2 answer file. Thank you.

Round 2

Reviewer 1 Report

The revised article “Inactivation of human norovirus G.4 and Vibrio parahaemolyticus in the sea squirt (Halocynthia roretzi) by floating electrode-dielectric barrier discharge plasma” is more understandable and clearer than the previous version. However, there are still some issues in the manuscript that need to be modified.

Introduction

Line 68

The references number 6 and 7 it is not suitable here move it together with the reference 9

Line 78

Similar the references number 12, 13 and 14 it is not suitable here (they have no data on korea raw fish products), move them above in line 77 after ….between June and October.

Materials and methods

Line 163 and 164. Explain NA or remove. Also extraction. It is sufficient write forward and reverse.

Table 1: In column of Sequence COG2R: 5’-TCG…..Remove COG2R

Results

Line 224….activity… the term is not suitable. Please change it. For example with ….trend.

Table 2 caption. Please change…. incubated sea squirt… with… spiked on sea squirt.

Table 3 is present 2 times.

Table 4 must be moved below the caption in line 284

Table 6 is present 2 times.

Line 299. “This” is correct? Please check it

Discussion

From lines 352 to 361. Living and dead virus is not suitable. Please change the terms with infectious or non-infectious.

Line 364: change sterilize with disinfect.

Back Matter

Line 424 Change Acknowledgements with Funding

Author Response

I marked the manuscript in red.

Reviewer 2 Report

It is suggested to summarize Conclusion and foccus on your own findings and your future plan

Author Response

I marked the manuscript in red.
